# Migration alters oscillatory dynamics and promotes survival in connected bacterial populations

Shreyas Gokhale [1], Arolyn Conwill [1], Tanvi Ranjan [2] & Jeff Gore [1]

Migration influences population dynamics on networks, thereby playing a vital role in scenarios ranging from species extinction to epidemic propagation. While low migration rates prevent local populations from becoming extinct, high migration rates enhance the risk of global extinction by synchronizing the dynamics of connected populations. Here, we investigate this trade-off using two mutualistic strains of *E. coli* that exhibit population oscillations when co-cultured. In experiments, as well as in simulations using a mechanistic model, we observe that high migration rates lead to synchronization whereas intermediate migration rates perturb the oscillations and change their period. Further, our simulations predict, and experiments show, that connected populations subjected to more challenging antibiotic concentrations have the highest probability of survival at intermediate migration rates. Finally, we identify altered population dynamics, rather than recolonization, as the primary cause of extended survival.

[1] Physics of Living Systems, Department of Physics, Massachusetts Institute of Technology, Cambridge, MA 02139, USA. [2] John A. Paulson School of Engineering and Applied Sciences, Harvard University, Cambridge, MA 02138, USA. These authors contributed equally: Shreyas Gokhale, Arolyn Conwill. Correspondence and requests for materials should be addressed to J.G. (email: gore@mit.edu)

Spatially extended populations can be distributed hetero- geneously, often in the form of a network of relatively dense population patches connected to each other by migration[1,2]. In macroecology, population networks are pervasive in a variety of social and ecological settings such as human settlements con- nected by transportation routes[3], oceanic islands connected by migration[4], and faraway plant populations connected by seed dispersal[5,6]. Migration patterns shape the population dynamics on these networks[7–9] and can have a profound impact on population stability and persistence[5,10,11].

One major focus of both theoretical work and field studies on connected populations has been conservation ecology, particu- larly in the context of endangered species[12]. In general, dis- connected networks or fragmented habitats are considered undesirable because isolated populations are susceptible to extinction due to demographic stochasticity or environmental fluctuations[13]. In the presence of migration, individuals from neighboring populations can stabilize an endangered popula- tion[14]. In addition, migration can counteract stochastic extinction of a local population by creating a metacommunity with a larger total population size[10]. Furthermore, migrants can repopulate nearby patches that have gone extinct[13,15]. The concept that migration can prevent permanent population collapse has led to the construction of conservation corridors that are intended to prevent local extinctions by facilitating movement between pre- viously unconnected habitat patches[16].

However, excessive migration can lead to synchronization of population dynamics in connected habitat patches[12,17–21]. In harsh environmental conditions, synchronization can enhance the risk of global stochastic extinction during periods of collective decrease in population size[12], since patches effectively merge into one large population in the limit of strong coupling[22]. Accord- ingly, a number of computational models predict that inter- mediate migration rates optimize species persistence time and extend metapopulation extinction time, mainly due to recoloni- zation events following local extinction[23–26]. Such recoloniza- tion-mediated enhancement in survival has been observed experi- mentally in metapopulations of plants[5], plants and predatory mites[27], fruit flies[10], and ciliate predator-prey systems[15,28].

Apart from enabling recolonization, migration can also perturb population dynamics, potentially giving rise to another mechan- ism of enhanced survival. Populations exhibiting deterministic oscillations are ideally suited to studying migration-induced perturbations in population dynamics. In particular, time course data enables both quantification of perturbations and identifica- tion of extinction and recolonization events, making it possible to ascertain the relative importance of each of these survival mechanisms. While theoretical work on nonlinear maps[29] and on predator-prey systems[30] has linked perturbed within-patch population dynamics to enhanced survival, experimental valida- tion is lacking owing largely to difficulties in obtaining high- quality time series data.

Here, we develop a bacterial model system composed of two connected oscillating populations to elucidate the impact of migration on population dynamics (Fig. 1a). Our approach is similar to previous studies[31–37] that have employed simple microbial populations in a laboratory setting to test ecological theories. Although it forgoes the complexity of natural population networks, our system is ideal for systematically exploring the relationship between survival and perturbed population dynamics because it allows us to precisely control the migration rate as well as environmental conditions. We also use a mathematical model of antibiotic degradation by bacteria to better understand the effects of migration between patches on our experimental system.

In this work, we first quantify the onset of synchronization in benign environmental conditions, in which individual populations

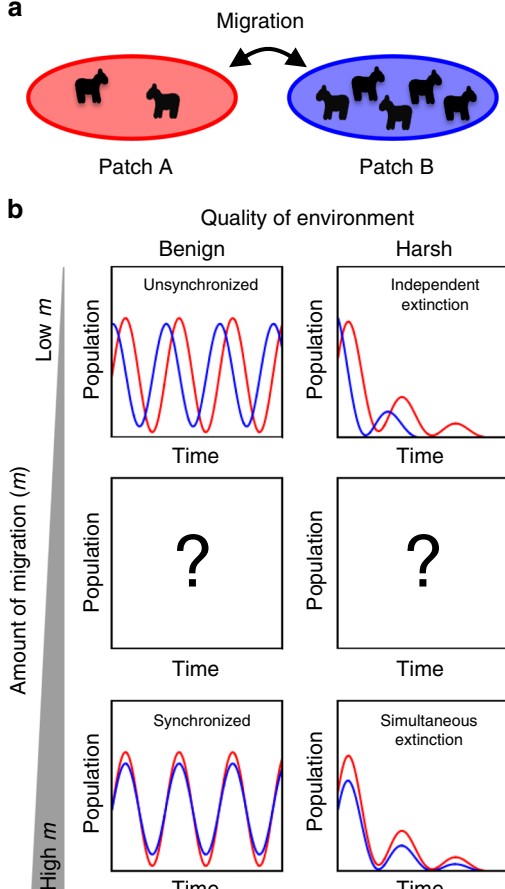

**Fig. 1** The cartoon illustrates possible qualitative effects of migration on population dynamics of a species in benign as well as harsh environmental conditions. **a** The population dynamics of two patches can be coupled via migration. **b** Under benign conditions, low migration rates are insufficient to couple population dynamics on the two patches (top left). On the other hand, high migration rates lead to synchronization, rendering the two patches equivalent (bottom left). An open question concerns how the population dynamics respond to intermediate migration rates (center left). In harsh conditions, population patches can become extinct (top right) and high migrations rate could lead to simultaneous extinction (bottom right). At intermediate migration rates, we investigate whether altered population dynamics could potentially lead to longer survival times (center right)

exhibit stable oscillations over the duration of the experiment (Fig. 1b). More importantly, we show that en route to synchroni- zation, the system goes through a series of qualitatively distinct oscillatory dynamics that are not observed in the absence of migration. We further show that these new dynamics enable populations to endure longer in harsh environments, as evidenced by the increase in survival times for moderate levels of migration. We emphasize that intermediate migration rates can lead to dif- ferent ecological outcomes, even though the migration rate is below the level necessary for the onset of synchronization (Fig. 1b). In a broader context, our results on two connected bacterial popula- tions can be viewed as the first step in a bottom-up approach aimed at probing the role of migration in the population dynamics of more complex networks.

## Results

**Experimental setup**. In order to quantify effects such as syn- chronization in an experimental model system, we use a bacterial

cross-protection mutualism that exhibits robust oscillatory population dynamics[31]. The system is comprised of two strains of *Escherichia coli* that protect each other from antibiotics in the environment by producing resistance enzymes. One strain (AmpR) is resistant to the antibiotic ampicillin, and the other strain (ChlR) is resistant to the antibiotic chloramphenicol (Fig. 2a). Previous work has already demonstrated that a co-culture of AmpR and ChlR exhibits robust limit cycle oscillations as a function of time over a broad range of antibiotic concentrations, when subjected to serial daily dilutions into fresh media and antibiotics[31].

In the serial daily dilution scheme employed in our experiments (Fig. 2b), we propagated co-cultures of the AmpR and ChlR strains in ~24 h growth-dilution cycles in 96-well plates under well-mixed conditions in the presence of LB media, ampicillin, and chloramphenicol. At the end of each growth cycle, we measured the total population density using spectrophotometry and the relative proportion of AmpR and ChlR cells using flow cytometry (see Methods). Next, we diluted the co-cultures by a factor of 100 into fresh media containing antibiotics, and subjected them to another growth cycle for ~24 h. Upon repeating these daily growth-dilution cycles for 15 days, we observed that the population density of AmpR (purple) as well as ChlR (green) cells oscillates with a period of 3 days (Fig. 2c, left panel). We note that the ratio of AmpR cells to ChlR cells constitutes an appropriate characterization of the state of the population because it also exhibits period-3 oscillations, with the oscillation amplitude spanning four orders of magnitude (Fig. 2c, right panel).

In order to examine the effects of migration, we studied pairs of co-cultures and coupled them via transfer of cells between the two patches of each pair. Figure 3a depicts the growth-migration-dilution scheme employed in our experiments. The scheme is similar to the growth-dilution scheme discussed earlier (Fig. 2b), with two important additions. First, we consider two co-cultures instead of one, which we label as habitat patches A and B. The second crucial addition is the migration step, which occurs after growing the co-cultures for 24 h and measuring the total population density and relative abundances of the strains, but before dilution into fresh media. In this step, we transfer a fraction $m$ of the cells from each co-culture into the other (corresponding to a fixed volume). The mixed co-cultures are then diluted into fresh media with antibiotics and grown again for 24 h, as done previously for individual co-cultures. This migration scheme allows us to vary the migration rate over several orders of magnitude, enabling us to experimentally probe the effect of migration on population dynamics, in a systematic manner.

**Synchronization of oscillations in benign conditions.** Since migration is known to lead to synchronization in a variety of systems[12,18–21], we first quantified the minimum migration rate necessary for the onset of synchronization under benign environmental conditions (10 μg/ml ampicillin, 8 μg/ml chloramphenicol), in which the system exhibits stable oscillations (Fig. 2c). For each migration rate, we performed daily growth-migration-dilution experiments with three initial relative abundances of AmpR to ChlR cells for each of two biological replicates, i.e., a total of six replicate pairs of co-cultures. While synchronization can sometimes be observed even at very low migration rates (and even at $m = 0$ when the two populations happen to oscillate in phase without coupling), all replicates showed synchronous dynamics for $m \geq 0.2$ (Fig. 3b). This strongly suggests that the onset of complete synchronization lies in the region $0.1 < m \leq 0.2$.

While synchronization is certainly an important effect, it is not clear whether it is the only effect that migration has on

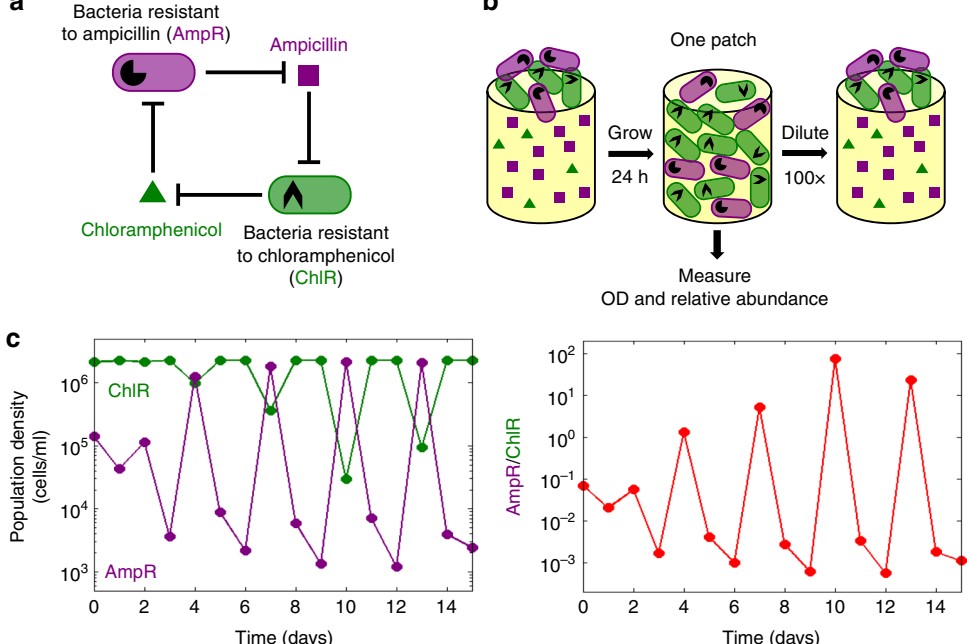

**Fig. 2** A bacterial cross-protection mutualism serves as a model system to study migration-induced synchronization of population oscillations. **a** Diagram depicting the mutualistic interaction between ampicillin resistant (AmpR) and chloramphenicol-resistant (ChlR) cells. AmpR cells protect ChlR cells by enzymatically deactivating ampicillin, whereas ChlR cells protect AmpR cells by deactivating chloramphenicol. **b** Schematic illustration of the experimental growth-dilution scheme for growing isolated co-cultures in the absence of migration. Each day, cells are grown for 24 h and then diluted by a factor of 100 into fresh media and antibiotics. The total cell density as well as relative proportions of AmpR and ChlR cells are measured after 24 h of growth, before the dilution step. **c** Isolated co-cultures exhibit period-3 oscillations in the density of AmpR and ChlR cells (left panel) as well as in the ratio of AmpR cells to ChlR cells (right panel) under benign conditions. This experimental condition corresponds to 10 μg/ml of ampicillin and 8 μg/ml of chloramphenicol

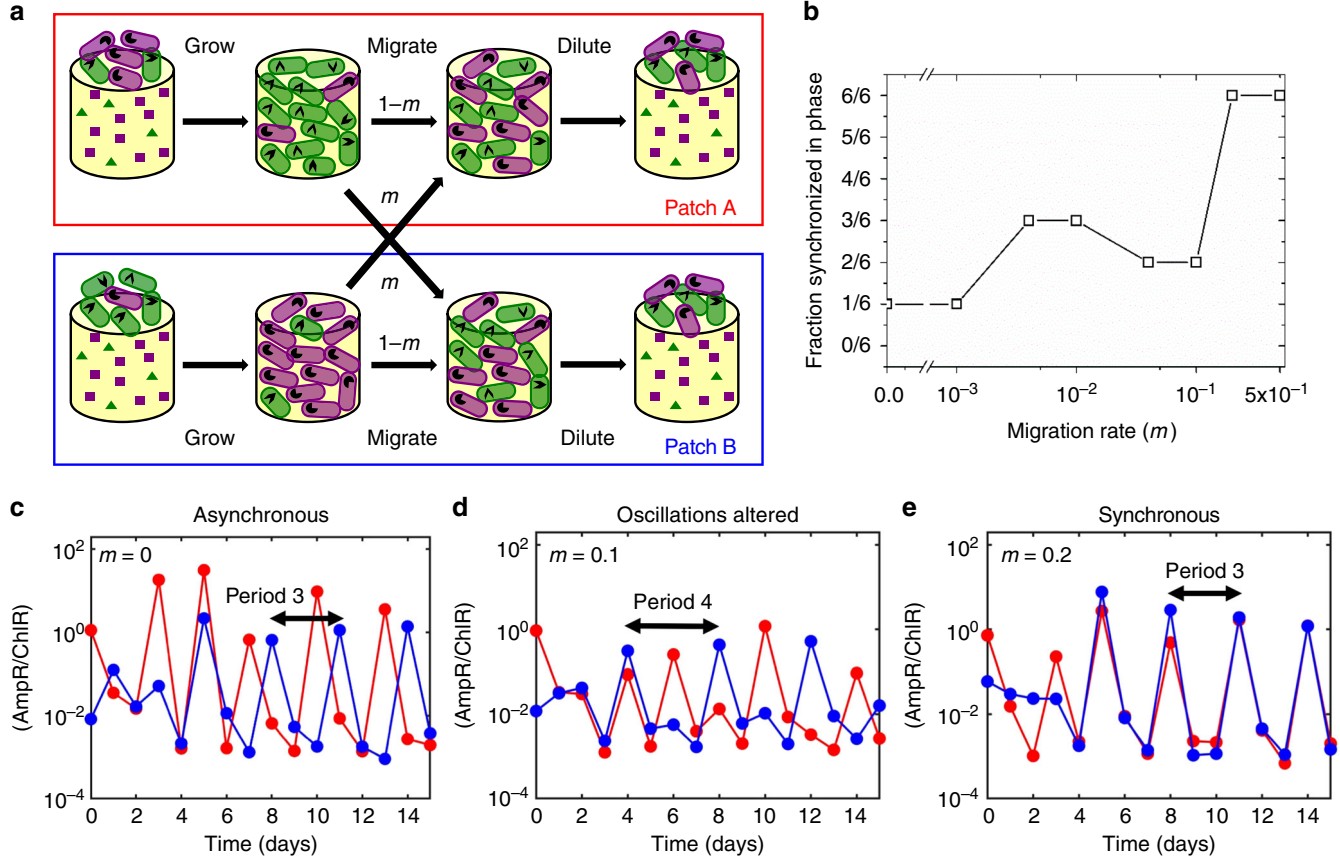

**Fig. 3** Increased migration rate leads to altered population dynamics and ultimately to synchronization. **a** Schematic illustration of the growth-migration-dilution scheme employed in the experiments with two connected bacterial populations. The two patches A (red box) and B (blue box) correspond to two distinct co-cultures of AmpR and ChlR cells. The migration rate is denoted by m. **b** The fraction (out of six replicates) of connected pairs of co-cultures synchronized in phase as a function of the migration rate in benign environmental conditions. **c–e** Representative time series for the ratio of AmpR cells to ChlR cells in patches A (red data series) and B (blue data series) for m = 0 (**c**), m = 0.1 (**d**), and m = 0.2 (**e**), showing asynchronous period-3 oscillations, disturbed oscillations with various periods and synchronous period-3 oscillations, respectively. In **b–e**, the experimental condition corresponds to 10 µg/ml of ampicillin and 8 µg/ml of chloramphenicol

population dynamics. In particular, it is worth investigating whether there are additional qualitative differences in population oscillations at intermediate migration rates. To this end, we took a closer look at the population time series data as a function of increasing migration rate. It is evident that our coupled co-cultures exhibit period-3 oscillations in the absence of migration (m = 0, Fig. 3c) as well as very high migration rates (m = 0.2, Fig. 3e), the difference being that the oscillations are asynchronous in the former case and synchronous in the latter. However, the period-3 oscillations are perturbed at intermediate migration rates, and we see signatures of other periods, such as period-4 oscillations at m = 0.1 (Fig. 3d). The perturbed oscillations may also reflect the presence of long transients[38]. Interestingly, perturbed oscillations appear to be quite common for m = 0.04 and m = 0.1 but are not observed at very low or very high migration rates (Supplementary Figure 1).

**Mechanistic model of population dynamics**. The data in Fig. 3c–e demonstrate that migration not only synchronizes population dynamics but also alters them in a qualitative manner. To better understand the sequence of experimental outcomes with increasing migration rate, we turned to an ordinary differential equation based mechanistic model that simulates antibiotic degradation and cell growth. This model was developed in the context of isolated co-cultures and has been shown to reproduce

the observed period-3 limit cycle oscillations over a reasonably broad parameter regime[31]. The model has two variables $N_1$ and $N_2$ corresponding to the population densities of AmpR and ChlR, respectively, and two variables $A_1$ and $A_2$ corresponding to the concentrations of the antibiotics ampicillin and chloramphenicol, respectively. Over the 24 h growth period, the population densities and antibiotic concentrations change with time according to the following equations:

$$\frac{dN_1}{dt} = \gamma_1(A_2)N_1\left(1 - \frac{N_1 + N_2}{K}\right) \quad (1)$$

$$\frac{dN_2}{dt} = \gamma_2(A_1)N_2\left(1 - \frac{N_1 + N_2}{K}\right) \quad (2)$$

$$\frac{dA_1}{dt} = \frac{-V_{max}A_1}{K_m + A_1}N_1(t = 0) \quad (3)$$

$$\frac{dA_2}{dt} = -c_2 A_2 N_2 \quad (4)$$

Here, we assume that bacterial growth is logistic, with antibiotic concentration-dependent growth rates $\gamma_1(A_2)$ and $\gamma_2(A_1)$. The two strains exhibit neutral resource competition, as reflected in

the combined carrying capacity $K$. Ampicillin degradation is assumed to obey Michaelis-Menten kinetics. In reality, ampicillin is degraded by β-lactamase[39,40] molecules produced by AmpR cells during their growth as well as the free ones carried over from the previous day. Our model assumes that the dominant contribution comes from the β-lactamases carried over from the previous day. Since the number of enzyme molecules carried over is proportional to the number of AmpR cells present at the beginning of the day, the equation for $A_1$ contains the initial density of AmpR cells $N_1(t=0)$ rather than their instantaneous density $N_1(t)$. These features of the model are supported by our previous findings that there is significant beta-lactamase activity in the media transferred from the previous day[41] and that the oscillatory population dynamics in our model of the mutualism most closely reflect those observed in experiments when we use the initial AmpR cell density to determine the ampicillin deactivation rate. Since chloramphenicol degradation is intracellular[39], the degradation rate of chloramphenicol is taken to be proportional to the density of ChlR cells. While our mathematical model explicitly considers the inoculum effect due to enzymatic degradation of antibiotics, it does not include other potential sources of an inoculum effect[42–44], as they were not necessary to recapitulate the dynamics observed in experiments. Finally, since AmpR cells are sensitive to chloramphenicol, and ChlR cells are sensitive to ampicillin, the growth rates of the two strains are proportional to the concentration of the antibiotic to which they are sensitive. These growth rates are given by:

$$\gamma_1(A_2) = \begin{cases} 0 & t < t_{\text{lag}} \\ \frac{\gamma_1^{\text{R}}}{1+A_2/I_{12}} & t \geq t_{\text{lag}} \end{cases} \quad (5)$$

$$\gamma_2(A_1) = \begin{cases} 0 & t < t_{\text{lag}} \\ -\gamma_2^{\text{D}} + \frac{\gamma_2^{\text{R}}+\gamma_2^{\text{D}}}{1+A_1/I_{21}} & t \geq t_{\text{lag}} \end{cases} \quad (6)$$

Here, we have also incorporated a lag phase, characterized by a time scale $t_{\text{lag}}$ over which the cells do not grow or die. Since chloramphenicol is a bacteriostatic antibiotic, i.e., it inhibits sensitive cell growth but does not cause the cells to die, the growth rate of AmpR cells approaches zero at high concentrations of chloramphenicol but never becomes negative (Fig. 4a, green curve). On the other hand, ampicillin is bactericidal, i.e., its presence can cause sensitive cells to die and we therefore model the growth rate of ChlR cells such that it is negative at high concentrations of ampicillin (Fig. 4a, purple curve). Numerical values and descriptions of all parameters used in the simulations are listed in Supplementary Table 1. In a typical simulation over a 24 h growth cycle, the cell densities saturate (Fig. 4b, top panel) and the antibiotics are deactivated (Fig. 4b, bottom panel). We note that the initial and final densities of AmpR and ChR can be substantially different, such that the ratio $N_1/N_2$ can vary substantially from day to day.

To gain insight into the experimentally observed changes in oscillatory dynamics with increasing migration rate, we implemented the growth-migration-dilution scheme in our simulations. The chief advantage of simulations is that they allow us to analyze the system's behavior over timescales that are much longer than those accessible in our experiments. The long timescales enable us to discern whether a given dynamical outcome corresponds to stable oscillations or transient dynamics. Simulations also facilitate a detailed characterization of oscillatory dynamics as well as the range of migration rates over which they are observed. In our simulations, we started with two patches A and B for which we numerically integrated the model equations over 24 h. To implement the migration and dilution steps, we mimicked the experimental protocol by resetting the initial population densities and antibiotic concentrations in the two patches for the next day of growth in accordance with the dilution factor of 100 and the migration rate $m$. The system's dynamical outcomes are best summarized in the form of a bifurcation diagram as a function of the migration rate (Fig. 4c). In this bifurcation diagram we have plotted the unique values attained by the population, i.e., the ratio of AmpR to ChlR cells in patch A over the last 50 days of a simulation consisting of 1000 daily dilution cycles.

As expected, we observe a regime of period-3 oscillations at very low migration rates (Fig. 4c, blue region) as well as high migration rates (Fig. 4c, red region). Moreover, these oscillations are asynchronous at very low migration rates and synchronous at high migration rates (Fig. 4c, insets corresponding to blue and red regions. Also see Supplementary Figure 2 for the migration rate dependence of the probability of synchronization.). Quite remarkably, the bifurcation diagram also contains a regime of period-4 limit cycle oscillations (Fig. 4c, orange region and inset) at intermediate migration rates that is surrounded on both sides by narrow regions characterized by irregular dynamics and long transients (Fig. 4c, green regions). The transition from period-3 to period-4 dynamics occurs via an inverse period doubling cascade (Fig. 4c, $0.05 \leq m \leq 0.06$). Such inverse period doubling cascades are commonly observed in discrete two-patch population dynamics models such as coupled logistic maps or Ricker maps[29]. Although our model is based on ordinary differential equations, the daily dilution scheme imposes discreteness on the dynamics, and the observed bifurcation structure is analogous to that observed in coupled maps. The dynamics at intermediate migration rates in Fig. 4c are strikingly similar to the perturbed oscillations observed in our experiments (Fig. 3d and Supplementary Figure 1). Indeed, our simple model successfully captures the sequence of experimentally observed dynamical outcomes as a function of the migration rate. In particular, transitions between different dynamical regimes are in reasonable qualitative agreement with our experiments. Moreover, the presence of altered population dynamics in both experiments and simulations suggests that intermediate migration rates can indeed give rise to population dynamics that are distinct from those in the asynchronous and synchronous regimes.

**Intermediate migration rates enhance survival.** It is plausible that the disturbance of oscillations at intermediate migration rates may influence a population's viability in harsh environments, although it is not a priori obvious whether this influence would be harmful or beneficial. As mentioned earlier, it has been suggested that synchronization can enhance the risk of global extinction in harsh environments[12,18], which implies that high migration rates may have a deleterious impact on the probability of survival, but it is not clear if this effect is monotonic. To explore the effect of intermediate migration rates on survival probability, we simulated our model in harsh environmental conditions. For these simulations, a harsh environment corresponds to higher antibiotic concentrations (10 μg/ml ampicillin, 16 μg/ml chloramphenicol), where the model predicts that isolated populations go extinct deterministically in the absence of migration. Furthermore, we introduce 15% noise in the migration and dilution steps of our simulations, to mimic the stochastic fluctuations resulting from our experimental protocol. To quantify the impact of migration on survival, we generated probability distributions $P(\tau)$ of population lifetimes, i.e., the durations $\tau$ for which the populations survived for various $m$ (Fig. 5a). The distributions decay exponentially over a timescale that increases with migration rate for $m \lesssim 0.03$ and decrease with migration rate for $m \gtrsim 0.1$.

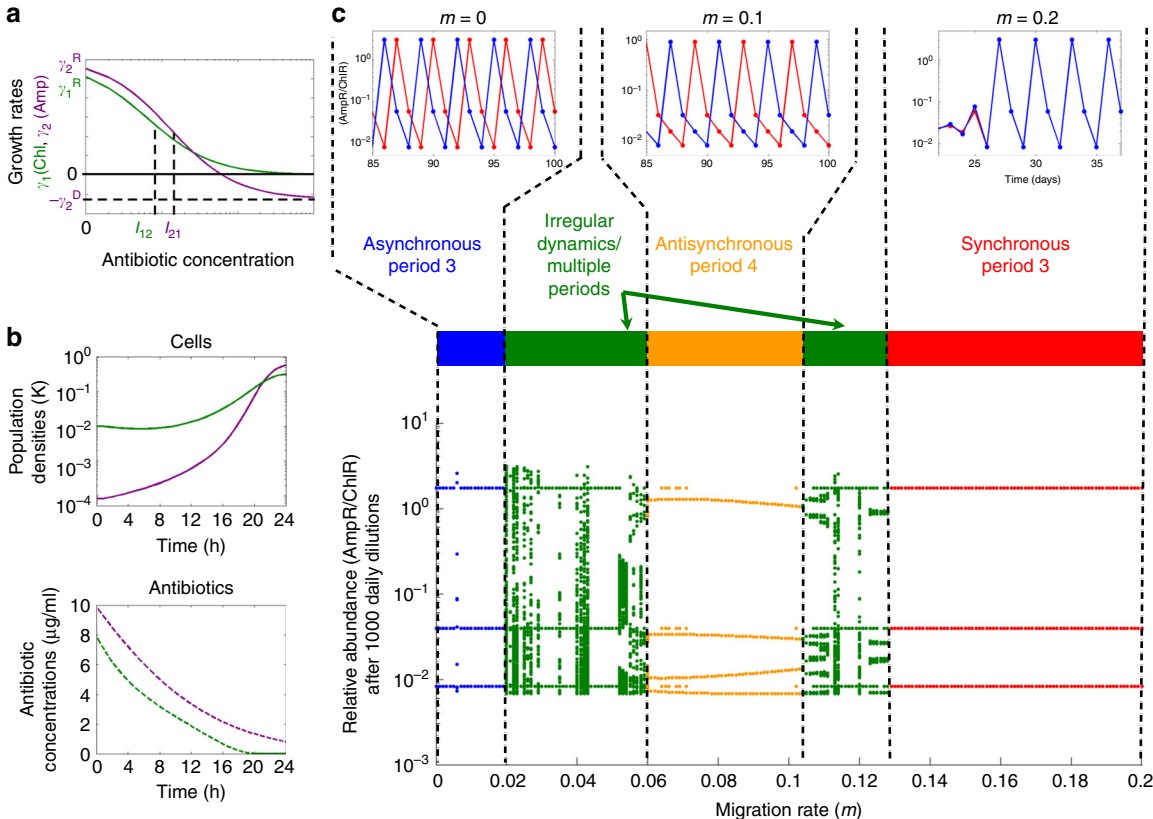

**Fig. 4** A mechanistic model of antibiotic degradation captures the experimentally observed sequence of dynamical outcomes. **a** Dependence of growth rates on the concentration of antibiotics. **b** Simulated ampicillin resistant (AmpR) and chloramphenicol resistant (ChlR) cell densities (top panel) and antibiotic concentrations (bottom panel) over the course of one 24 h growth cycle, shown in purple and green, respectively. **c** Bifurcation diagram for a simulation of two co-cultures in a benign environmental condition (10 μg/ml of ampicillin, 8 μg/ml of chloramphenicol) as a function of the migration rate $m$. Unique values of the subpopulation density ratio (AmpR/ChlR) attained by patch A at the end of the growth cycle over the last 50 days of a simulation with 1000 daily dilutions are plotted for each migration rate. The simulations are deterministic. The diagram captures the sequence of observed dynamical outcomes: asynchronous period-3 oscillations in the absence of migration, period-4 oscillations and irregular dynamics at intermediate migration rates, and synchronous period-3 oscillations at large migration rates. The insets in **c** show representative time series for $m = 0$, $m = 0.1$, and $m = 0.2$. Model parameters can be found in Supplementary Table 1

Interestingly, within a narrow intermediate migration regime ($0.04 \lesssim m \lesssim 0.07$), the survival time distributions have longer tails, which suggests that intermediate migration rates can result in enhanced survival as compared with isolated populations or strongly coupled ones. Overall, the variation of $P(\tau)$ with $m$ indicates that moderate levels of migration offer the best chance of survival in harsh conditions.

A more experimentally tractable measure of the effect of migration on survival in challenging environments is the fraction of populations that survive over a given period of time. Towards this end, we computed the probability of survival over 10 days. As expected from the variation in $P(\tau)$ with $m$ (Fig. 5a), the survival probability over 10 days changes non-monotonically with the migration rate (Fig. 5b). This qualitative trend occurs regardless of the duration over which we compute survival probability (Supplementary Figure 3). Moreover, the peak in survival probability occurs in the regime where we observe approximately exponential decay in $P(\tau)$, once again suggesting that moderate levels of migration perturb population dynamics in a manner that favors extended survival.

Motivated by the simulation results, we proceeded to test whether the predicted non-monotonicity of survival probability with $m$ is also observed in experiments. Accordingly, we performed growth-dilution-migration experiments with higher antibiotic

concentrations (10 μg/ml ampicillin, 16 μg/ml chloramphenicol) and measured the fraction of populations that survived over 10 days. As predicted by the simulations, the survival probability indeed shows a peak at intermediate migration rates (Fig. 5c; see Supplementary Figure 4 for population density time series). Further, the location of the maximum shows reasonably good quantitative agreement with the simulations. Collectively, these findings establish that moderate levels of migration promote population dynamics that extended survival in harsh environments.

Interestingly, the dynamics we observe for moderate migration rates resemble noisy antisynchronous period-2 oscillations (see Supplementary Figure 5 for bifurcation diagrams and a representative time series), and we see signatures of such oscillations in some of our surviving experimental populations as well (Supplementary Figure 6). Intuitively, antisynchronous dynamics could ensure that the population sizes in the two patches do not simultaneously become low, which averts the danger of global extinction. In previous computational studies[23,29,45,46], such antisynchronous dynamics have been widely recognized as a potential mechanism for survival and our experiments provide direct evidence in support of these numerical findings.

To investigate whether the complex antisynchronous dynamics observed in our model (Supplementary Figure 5) are chaotic, we

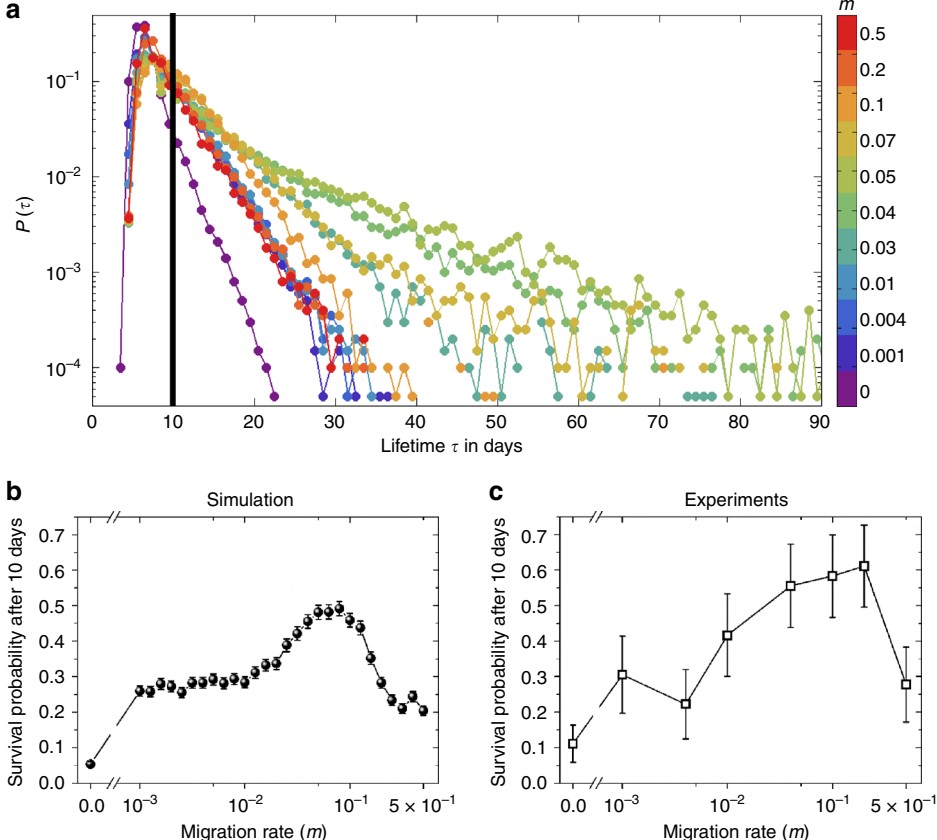

**Fig. 5** Moderate levels of migration help populations survive longer in harsh environments. **a** Simulated probability distributions of survival times of populations in a harsh environment (10 µg/ml of ampicillin, 16 µg/ml of chloramphenicol) for various migration rates. The distributions were generated from 6000 simulation runs with initial conditions distributed around the three phases of the period-3 oscillations observed in Fig. 4c. Connected patches were initialized in different phases to avoid minimize synchronization. $P(\tau)$ is defined as the fraction of initial conditions that survived for $\tau$ days. Different colors represent different migration rates, as indicated in the colorbar. The survival time distributions have longer tails at intermediate migration rates. The black vertical line indicates the threshold (10 days) used for calculating the survival probability. This threshold was chosen to match the duration of the experiments. **b** Simulated probability of survival after 10 days in the harsh environment as a function of the migration rate. **c** Experimentally measured survival probability after 10 days as a function of the migration rate from 18 replicate pairs (3 biological replicates × 6 technical replicates) of co-cultures. Both **b** and **c** exhibit a maximum at intermediate migration rates, demonstrating that moderate amounts of migration help populations to survive longer in harsh environments. The error bars in **b** and **c** are standard errors of proportion

computed[47] the largest Lyapunov exponent from simulated trajectories of the log ratio of AmpR cells to ChlR cells in a given patch, for $0.02 \leq m \leq 0.06$. We find that chaotic dynamics are indeed present within this regime, as evidenced by positive values for the largest Lyapunov exponent for several migration rates within the range $0.02 \leq m \leq 0.042$ (Supplementary Figure 7). Since we do not observe period doubling, it is possible that the quasiperiodic route to chaos underlies the chaotic dynamics observed here. The quasiperiodic route has been observed in a system of two linearly coupled logistic maps[48] as well as other population dynamics models[49,50]. While a detailed analysis of the routes to chaos is outside the scope of the present work, we reiterate that the discreteness imposed by daily dilutions may render the dynamics of our system analogous to those observed in coupled nonlinear maps. While the migration rate that maximizes survival probability ($m \sim 0.05$) is slightly larger than the regime over which we observe chaos, the data in Fig. 5b and Supplementary Figure 7 show that antisynchronous chaotic dynamics contribute to enhanced survival in our model. Given the close agreement between our experiments and simulations, it is tempting to speculate that chaotic dynamics may also help our

experimental bacterial populations to survive longer in harsh conditions.

## Discussion

Here, we have shown that two oscillating bacterial populations can synchronize when coupled sufficiently strongly via migration. Furthermore, we have demonstrated that it is possible for the migration rate itself to determine the period of oscillation. In particular, our experimental system exhibits limit cycle period-3 oscillations in the absence of migration. However, we observed disturbances in these dynamics in the presence of migration that were consistent with the period-4 oscillations predicted by the model at intermediate migration rates (Fig. 3d). This finding represents empirical evidence that the characteristics of population oscillations observed in natural microbial communities may not simply be a result of intrinsic inter- or intra-species interactions, but may also be a consequence of spatial structure and migration.

Since migration can perturb population dynamics, it also has the potential to influence the survival of a population in a challenging environment. For instance, we found that populations were most likely to survive the duration of the experiment at

intermediate migration rates (Fig. 5). The presence of a maximum in survival probability at intermediate migration rates is potentially relevant in conservation biology and epidemiology in that controlling migration might lead to desired outcomes such as population stability or disease eradication.

In field ecology and metapopulation theory, recolonization of habitat patches after local extinction events is thought to be a major contributor to extended survival in harsh environments[13], a claim supported by recent experiments on protists[15]. However, we observed relatively few instances of such recolonization (Supplementary Figure 8), implying that recolonization is not the major cause of extended survival in our system. Nevertheless, in a more spatially extended population with a larger number of habitats, it may be possible that recolonization plays a more significant role in enabling populations to survive in challenging conditions.

Finally, we note that in challenging environments, evolutionary rescue can also lead to population recovery[51]. In our system, evolutionary rescue may occur either via enhanced antibiotic tolerance through the evolution of lag time mutants[52] or via increased drug resistance in one of the strains. Indeed, we did observe a few cases in which ChlR cells developed a higher resistance to ampicillin, particularly at high migration rates in the harsh environment (10 μg/ml ampicillin, 16 μg/ml chloramphenicol) (Supplementary Figure 9). Interestingly, we found no evidence of such evolution in extremely harsh environments (10 μg/ml ampicillin, 20 μg/ml chloramphenicol), in which all populations became extinct within 7 days (Supplementary Figure 10). These observations suggest that the optimal conditions for evolution of additional antibiotic resistance represent a tradeoff between selection pressure, which is primarily determined by the environment, and survival time, which can be influenced by the migration rate. In addition, the effective population size may also play a role in guiding the course of evolution in that stronger coupling may lead to a larger population size and thus more genetic diversity, but may also lead to reduced survival time. Collectively, our findings highlight the importance of spatial structure and its impact on bacterial population dynamics in the context of the evolution of antibiotic resistance[53,54]. It is evident that cross-protection can render a population resistant to multiple drugs without any individual strain evolving resistance to more than one drug[31]. Furthermore, similar to how drug tolerance can promote the evolution of resistance[55], it is possible that cross-protection, particularly in combination with spatial structure and migration, could provide time for strains to evolve multidrug resistance de novo.

In the present study, we found that moderate amounts of migration between two coupled bacterial populations can significantly perturb population oscillations and enhance survival in harsh environments (Fig. 3d and Fig. 5c). Given that a simple setup can have significant ecological consequences, it would be interesting to extend this approach to a larger number of connected populations. For example, in addition to the migration rate, the network topology can play an important role in governing synchronization[56,57]. The network topology also allows exotic forms of partial synchronization such as phase clusters and chimeras[58–60], whose relevance to ecological systems is hitherto unexplored. The effects of asymmetric migration rates, spatial expansion, and environmental conditions are also worthy of exploration[45,61–63].

More generally, our experiments suggest that it is worthwhile to explore how concepts from macroecology and network theory apply to microbial systems. Despite extensive work in theoretical ecology and macroecology, relatively little attention has been dedicated to examining the implications of migration on microbial communities, which can exhibit rich population dynamics over spatially extended environments. Spatially fragmented yet dispersed microbial communities can be found in the ocean on organic particulate matter called marine snow[64], in the ground on soil grains, and on different human body sites, suggesting that the impact of migration on the ecology of these microbial ecosystems may have implications on topics ranging from the global carbon cycle[65] to human health. Given that that there are notable differences between microbes and larger organisms (e.g., relatively large population sizes with fewer stochastic fluctuations), further work is necessary to translate findings from theoretical ecology and macroecology to microbial ecology.

## Methods

**Strains.** The strains used are identical to those in our previous paper[31]. Briefly, the chloramphenicol-resistant strain ChlR is an *E. coli* DH5α strain transformed with the pBbS5c-RFP plasmid[66] which encodes a gene for chloramphenicol acetyltransferase (type I) enzyme as well as a gene for monomeric red fluorescent protein (RFP). The plasmid pbBS5c-RFP was obtained from Jay Keasling (University of California, Berkeley, CA) via Addgene (plasmid 35284)[66]. The ampicillin-resistant strain is an *E. coli* DH5α strain transformed with a plasmid encoding a gene for the β-lactamase enzyme (TEM-1) and a gene for enhanced yellow fluorescent protein (EYFP).

**Experiments.** Initial monocultures of our strains were grown for 24 h in culture tubes containing 5 ml LB supplemented with antibiotic for selection (50 μg/mL ampicillin and 25 μg/mL chloramphenicol for AmpR and ChlR, respectively) at 37 °C and shaken at 250 rpm. The following day, 200 μL of co-cultures of the two strains were grown at varying initial population fractions in LB without antibiotics. For synchronization experiments, the initial ratios of AmpR cells to ChlR cells were chosen such that the probability for paired co-cultures to be in the same phase of oscillation in the absence of migration (m = 0) was low. Subsequently, serial migration-dilution experiments were performed in well-mixed batch culture with a culture volume of 200 μL. In each cycle, co-cultures were grown for 24 h in LB media supplemented with the antibiotics ampicillin and chloramphenicol. During growth, cultures were shaken at 500 rpm at a temperature of 37 °C. At the end of the growth cycle, we measured the optical density (OD) at 600 nm and prepared flow cytometry samples by diluting 5 μL of each grown co-culture by a factor of 1600 into phosphate buffer (PBS, Corning 21-040-CV). To perform the migration step, we pipetted fixed volumes of each co-culture within a connected pair into the other at the end of each growth cycle; subsequently, these co-cultures were diluted by a factor of 100 into fresh LB media and antibiotics. Growth medium was prepared by using BD's Difco TM LB Broth (Miller) (catalog no. 244620). Ampicillin stock was prepared by dissolving ampicillin sodium salt (Sigma-Aldrich catalog no. A9518) in LB at a concentration of 50 mg/mL. The solution was filter sterilized, stored frozen at −20 °C, and thawed before use. Chloramphenicol stock was prepared by dissolving chloramphenicol powder (Sigma-Aldrich catalog no. C0378) in 200 proof pure ethanol (KOPTEC) at a concentration of 25 mg/mL. This solution was filter sterilized and stored at −20 °C. Prepared 96-well plates of media supplemented with antibiotics were stored at −80 °C, thawed one day prior to inoculation at 4 °C and warmed for 1 h at 37 °C immediately before inoculation.

**Measurement and data analysis.** At the end of each growth cycle, we took spectrophotometric (Thermo Scientific Varioskan Flash at 600 nm) measurements, which serve as a proxy for the total population size. We converted the OD measurement to CFU/μl based on a calibration curve obtained from counting colonies on LB and agar plates originating from cultures at various cell densities. We also took flow cytometry (Miltenyi Biotec MACSQuant VYB) measurements of the cultures to determine subpopulation sizes (see Supplementary Figure 11 for a representative pseudocolor plot exemplifying our gating strategy). We consider two populations to be synchronous if the peaks of their oscillation occur at the same time over the last two oscillation cycles (the final 40% of the experiment), to reduce the influence of transients as well as gather sufficient statistics. Data analysis was performed using a combination of Matlab and Origin. Simulations were performed using Matlab. Flow cytometry data were analyzed using the Python package FlowCytometryTools[67].

**Code availability.** All data analysis code is available upon request.

## Data availability

Data are available upon request. A Reporting Summary for this Article is available as a Supplementary Information file.

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

## Acknowledgements

This work was primarily supported by NIH Grant R01 GM102311-01 and National Science Foundation CAREER Award PHY-1055154. The laboratory acknowledges support from the Pew Scholars in the Biomedical Sciences Program Grant 2010-000224-007, NIH R00 Pathways to Independence Award GM085279-02, Sloan Foundation Fellowship BR2011-066, the Allen Distinguished Investigator Program, and NIH New Innovator Award DP2. S.G. was supported by a Human Frontier Science Program cross-disciplinary postdoctoral fellowship. We also thank members of the Gore Lab for helpful discussions.

## Author contributions

A.C. and J.G. designed the research; S.G. and T.R. performed the experiments; S.G. and A.C. developed models; S.G. and A.C. analyzed data; and S.G., A.C., and J.G. wrote the paper.

## Additional information

**Competing interests:** The authors declare no competing interests.

