## [Peer Review File · Nature Communications]

Reviewers' Comments:

Reviewer #1:

Remarks to the Author:

The manuscript by Gokhale and colleagues investigates the impact of migration on the population dynamics of connected populations. Using experiments on mutualistic bacterial strains along with mathematical models, they show that introducing migration alters and—for sufficiently high migration rates—eventually (phase) synchronizes the oscillations of connected populations. A detailed mechanistic model captures the experimental behavior and points to a rich bifurcation structure as migration rate is increased. In particular, the model predicts and experiments confirm that intermediate migration rates maximize population survival in harsh environments, here corresponding to high antibiotic concentrations.

The paper is interesting and presents a compelling combination of theoretical and experimental work. A particular strength of the paper is that it leverages (relatively) simple, quantitative experiments to address potentially broader ecological questions. The experiments are well designed and innovative—for example, the dilution scheme allows the authors to simulate migration rates over several orders of magnitude without requiring complex flow systems. The model is simple enough to highlight the important features of the experiment yet makes novel and non-trivial predictions. Some of the findings—e.g. the optimality of intermediate migration rates for population survival—are based on mechanistic models specific to this microbial system, but one can imagine generalized models of mutualism that exhibit similar dynamics. Therefore, I believe that the results would interest not only ecologists, but also microbiologists and physicists/applied mathematicians studying dynamical systems. In addition, I think it's important to note that enzymatic degradation is a common mechanism of drug resistance. Therefore, while the system is—by construction—a simplified “toy” ecosystem, the results underscore the importance of spatial structure and population heterogeneity in fully understanding the spread of resistance.

I enjoyed the paper and consider it very well done. I do, however, have a few points for the authors to consider that might improve this or future work. Most questions could probably be addressed using the model. I don't intend for these suggestions to be mandatory, but they may further clarify the conclusions and/or broaden the appeal of the work for certain readers.

-- The richness of the bifurcation diagram from the model (figure 4c) makes me wonder if the dynamics for intermediate migration rates may have a chaotic component. While it is not uncommon to see chaos in ecological models (particularly discrete time seasonal models), I'm not aware of many cases where the chaos underlies important qualitative dynamics. I think it would be extremely interesting if, for example, the optimal migration rate (as in Figure 5) appeared in a chaotic or near chaotic regime. I recognize that establishing chaos in the experimental system would be very difficult, but it might not be so hard to look for chaos around the optimal point in the model (using, for example, Lyapunov exponents or similar). If it's there, it would be quite interesting and provide a biologically “useful” version of chaos: namely, harnessing the beneficial effects of migration without incurring the drawbacks of complete synchronization.

-- As I understand it, the dilutions that mimic migration involve transferring small liquid volumes that include both cells and non-cell material (e.g. molecules of each drug, nutrients, etc) between different wells. In most cases, I would imagine that the effects of the non-cell material are basically negligible, at least for very small transfer volumes (relative to total well volume). However, there are some cases where it could matter. For example, the authors' model suggests that the degradation due to beta-lactamase is dominated not by growing cells, but by the enzyme remaining from the previous transfer (at time 0). This leads me to wonder: are there other possible chemical species in this transfer volume

that might impact the dynamics? For example, is it possible that the drug concentration in each transfer volume makes a nontrivial contribution to the overall drug concentration? If so, might the experiments with higher migration rates (i.e. more transfers or larger transfer volume) have systematically different drug concentrations than those with lower migration rates? Such an effect could be particularly strong if, for example, the drugs exhibit strong synergy or strong antagonism, so that even small changes in the concentration of one drug could dramatically alter the cellular response. I use drug concentration as an example, but the same applies to any environmental variable (e.g. pH or nutrient level) that might be present at sufficiently high concentration to impact growth and vary systematically for different migration rates. In short, what I'm getting at is this: how confident are you that the dominant effect of changing migration rates is truly due to the migration of *cells* to different habitats (rather than the tag-along "migration" of other molecules, etc., in the environment)?

-- Many antibiotics are known to exhibit a strong inoculum effect (IE), where the initial density of cells modulates the inhibitory effects of the drug. You implicitly account for a cross-species IE in the model by modeling the impact of enzyme degradation. So, for example, the effect of Amp on the Chl-R cells will obviously depend on the density of Amp-R cells (or more directly, beta-lactamase enzyme) in the culture. In addition, a monoculture of Amp-R cells would be expected to exhibit a strong IE to ampicillin, and your model would capture that. But it's also possible that the Chl-R cells exhibit an IE when exposed to ampicillin, even when cultured alone, so that Chl-R growth in ampicillin depends on the initial density. The same goes for the Amp-R cells exposed to Chl. Because the densities of the subpopulations can change dramatically in this experiment, is it possible that the IE is playing a significant role here? To my knowledge, drugs similar to ampicillin are fairly often associated with an IE; sometimes this is due to production of beta lactamases—and again, you've covered that case in your model—but it can also be due to other mechanisms and/or result in increased drug efficacy at higher densities (see, for example, www.ncbi.nlm.nih.gov/pmc/articles/PMC180193/ or <http://journals.plos.org/ploscompbiol/article?id=10.1371/journal.pcbi.1005098>). These other mechanisms could potentially impart a strong ampicillin IE (or "reverse IE") to the Chl-R cells that wouldn't show up in your model. On the other hand, recent work in E. coli showed no IE with Chl (www.ncbi.nlm.nih.gov/pmc/articles/PMC3472685/), so it may not be so important for that drug.

Very minor points:

-- I would suggest that the authors (slightly) extend their discussion about the potential application of these results to the spread of antibiotic resistance, where cooperation between resistant and sensitive cells is increasingly recognized as an important contributor to population dynamics. In addition to cooperation, the current work highlights the importance of spatial structure and population heterogeneity for understanding how resistance spreads in complex communities. While these topics have been the focus of a number of recent papers, they are sometimes underappreciated in the study of resistance. I know resistance is not the focus of the current work, but I think it would nevertheless interest readers who study resistance.

-- The authors use both "recolonization" and "re-colonization", depending on location.

Reviewer #2:

Remarks to the Author:

In general, the match between the theoretical model and the experimental observations is stunning, and the authors do a very good job describing their model and experiment. This work adds an

interesting new facet to the potential for intermediate migration rates to enhance persistence, not by the common route involving recolonization but by an alteration of dynamics that enhances the longevity of the population.

Much of the work that this paper builds upon focuses on theoretical models or experiments where population dynamics are not subject to serial dilution effects. This is obviously a cornerstone of the approach needed to undertake the experimental work and I don't think that this takes away at all from the strength of this paper. I do wonder however, how much the findings rely on the fact that the system undergoes a substantial reset with each dilution. Of course, the period three cycles likely require the serial dilutions to be present, but I wonder how sensitive the transition to the period 4 cycles is on the dilution factor. Playing with this parameter in the model may also help to demonstrate the mechanism giving rise to the period 4 cycles.

Along this line of reasoning, I think the authors could improve on their description of the mechanistic basis for the change in dynamical behavior. The switch from a period 3 cycle to a period 4 cycle due to a change in migration rate is very novel and interesting and something that other models based on coupled oscillators have not shown. It is in fact hard to understand how increasing migration could effectively slow cycles down; passive migration has strongest effect in patches with low density and helps populations to increase quickly from the zenith of their cycle. For predator-prey oscillations this is often referred to as the 'slow' part of the cycle. My suspicion here is that somehow migration is acting in some alternative manner, to slow down the fast part of the cycle.

From figure 3 panels c,d,e (or figure 4c) it really isn't clear why there is a difference in persistence times among the three parameter regimes due to the fact that the ratio of cell types is shown rather than absolute densities. I think showing the dynamics of the absolute densities would also be helpful so that differences in the dynamics (e.g. the minimum densities) would be obvious.

Throughout the paper the authors use the term unsynchronized. I find this term to be imprecise. My preference is to use the terms synchronous to represent positively correlated patch dynamics, asynchronous to represent patches with no correlation, and antisynchronous for negatively correlated patches. This language has been adopted by a number of papers (although some authors argue that antisynchrony is just a special case of synchrony).

It isn't clear what role the lag in growth has in the model, nor how the authors chose the lag value. Is this set by independent measurements, or was the parameter fit to best be able to describe the dynamics? Nowhere in the paper was the value of the lag presented.

In addition, it isn't clear why the dynamics of A1 would depend only on the initial state of N1 rather than dynamically on the instantaneous value. Given that the model can only be analyzed numerically, this seems a rather odd simplifying assumption with no real benefit.

In the low migration regime, is the antisynchronous state attractive, i.e. does the same dynamic setup regardless of the initial conditions or is it sensitive to initial conditions?

In the intermediate parameter range $0.02 < m < 0.06$ it is most likely that the dynamic is a form of quasiperiodic chaos since there is no obvious period doubling cascade taking place. This is typically generated by an interaction of oscillators with an irrational ratio of periods but I have to admit that I'm puzzled as to exactly how this type of dynamic sets up in this instance. It would be interesting to speculate on why the complex dynamics emerge here

Reviewers' comments:

Reviewer #1 (Remarks to the Author):

Comment 1: The manuscript by Gokhale and colleagues investigates in the impact of migration on the population dynamics of connected populations. Using experiments on mutualistic bacterial strains along with mathematical models, they show that introducing migration alters and—for sufficiently high migration rates—eventually (phase) synchronizes the oscillations of connected populations. A detailed mechanistic model captures the experimental behavior and points to a rich bifurcation structure as migration rate is increased. In particular, the model predicts and experiments confirm that intermediate migration rates maximize population survival in harsh environments, here corresponding to high antibiotic concentrations.

The paper is interesting and presents a compelling combination of theoretical and experimental work. A particular strength of the paper is that it leverages (relatively) simple, quantitative experiments to address potentially broader ecological questions. The experiments are well designed and innovative—for example, the dilution scheme allows the authors to simulate migration rates over several orders of magnitude without requiring complex flow systems. The model is simple enough to highlight the important features of the experiment yet makes novel and non-trivial predictions. Some of the findings—e.g. the optimality of intermediate migration rates for population survival—are based on mechanistic models specific to this microbial system, but one can imagine generalized models of mutualism that exhibit similar dynamics. Therefore, I believe that the results would interest not only ecologists, but also microbiologists and physicists/applied mathematicians studying dynamical systems. In addition, I think it's important to note that enzymatic degradation is a common mechanism of drug resistance. Therefore, while the system is—by construction—a simplified “toy” ecosystem, the results underscore the importance of spatial structure and population heterogeneity in fully understanding the spread of resistance.

I enjoyed the paper and consider it very well done. I do, however, have a few points for the authors to consider that might improve this or future work. Most questions could probably be addressed using the model. I don't intend for these suggestions to be mandatory, but they may further clarify the conclusions and/or broaden the appeal of the work for certain readers.

Response 1: We thank the reviewer for their positive assessment of our work and for highlighting the significance of our work in a broader context.

Comment 2: The richness of the bifurcation diagram from the model (figure 4c) makes me wonder if the dynamics for intermediate migration rates may have a chaotic component. While it is not uncommon to see chaos in ecological models (particularly discrete time seasonal models), I'm not aware of many cases where the chaos underlies important qualitative dynamics. I think it would be extremely interesting if, for example, the optimal migration rate (as in Figure 5) appeared in a chaotic or near chaotic regime. I recognize that establishing chaos in the experimental system would be very difficult, but it might not be so hard to look for chaos around the optimal point in the model (using, for example, Lyapunov exponents or similar). If it's there, it would be quite interesting and provide a biologically "useful" version of chaos: namely, harnessing the beneficial effects of migration without incurring the drawbacks of complete synchronization.

Response 2: The reviewer has raised a very interesting point. To establish whether our model exhibits chaos at migration rates in the vicinity of the optimal point in Figure 5B, we have computed the largest Lyapunov exponent from simulated time courses using the method developed by Wolf *et al.* (A. Wolf, J. B. Swift, H. L. Swinney and J. A. Vastano, *Physica D*, 16(3), 285-317, 1985) over the range $0.02 \leq m \leq 0.06$ for the harsh condition. We observe that chaotic dynamics are indeed present in this regime, as evidenced by the largest Lyapunov exponent showing positive values (Figure S6 in the revised manuscript, shown below).

Fig. S6: Largest Lyapunov exponent as a function of migration rate for the deterministic mechanistic model in the harsh environment (10 $\mu\text{g/ml}$ of ampicillin, 16 $\mu\text{g/ml}$ of

chloramphenicol). The algorithm only returns positive Lyapunov exponents, which indicate chaos. Points shown below 0 correspond to periodic or quasiperiodic dynamics. We computed the exponents from time series of the log ratio of AmpR cells to ChIR cells in a given patch. To minimize the effect of transients, time series were generated by simulating 10,000 daily growth-dilution-migration cycles.

We also observe long chaotic transients, complex limit cycles with non-trivial periods such as 6, 12 and 14 and possibly quasiperiodicity as well. Poincare recurrence plots for the log ratio of population densities of the strains AmpR and ChIR (i.e. $x = \log(\text{AmpR}/\text{ChIR})$) reveal the existence of a rich variety of attractors as this regime of migration rates is traversed (see figure below). We note that the migration rate corresponding to maximal survival probability ($m \sim 0.05$) is somewhat larger than the regime over which we observe chaos ($0.02 < m \leq 0.042$). The asymptotic dynamics for the optimal migration rate are either periodic or quasiperiodic, rather than chaotic. Moreover, the chaotic regime is interspersed with limit cycles that may correspond to periodic windows or frequency locking (see attractor for $m = 0.025$ in the figure below). Nonetheless, since chaotic transients as well as asymptotic chaotic dynamics are antisynchronous, they contribute to enhanced survival at intermediate migration rates.

Changes to the manuscript: We have added a new figure to the supplementary information (Fig. S6) showing the largest Lyapunov exponent as a function of migration rate in the harsh environmental condition. We have added a new reference for the method used to compute Lyapunov exponents (Ref. 47). We have also added the following text (Lines 311-327) describing our analysis.

“To investigate whether the complex antisynchronous dynamics observed in our model (Fig. S5) are chaotic, we computed the largest Lyapunov exponent from simulated trajectories of the log ratio of AmpR cells to ChIR cells in a given patch, using the algorithm described in (47), for $0.02 \leq m \leq 0.06$. We find

that chaotic dynamics are indeed present within this regime, as evidenced by positive values for the largest Lyapunov exponent for several migration rates within the range $0.02 \leq m \leq 0.042$ (Fig. S6). Since we do not observe period doubling, it is possible that the quasiperiodic route to chaos underlies the chaotic dynamics observed here. The quasiperiodic route has been observed in a system of two linearly coupled logistic maps (48) as well as other population dynamics models (49, 50). While a detailed analysis of the routes to chaos is outside the scope of the present work, we reiterate that the discreteness imposed by daily dilutions may render the dynamics of our system analogous to those observed in coupled nonlinear maps. While the migration rate that maximizes survival probability ($m \sim 0.05$) is slightly larger than the regime over which we observe chaos, the data in Fig. 5B and Fig. S6 show that antisynchronous chaotic dynamics contribute to enhanced survival in our model. Given the close agreement between our experiments and simulations, it is tempting to speculate that chaotic dynamics may also help our experimental bacterial populations to survive longer in harsh conditions.“

Comment 3: As I understand it, the dilutions that mimic migration involve transferring small liquid volumes that include both cells and non-cell material (e.g. molecules of each drug, nutrients, etc) between different wells. In most cases, I would imagine that the effects of the non-cell material are basically negligible, at least for very small transfer volumes (relative to total well volume). However, there are some cases where it could matter. For example, the authors' model suggests that the degradation due to beta-lactamase is dominated not by growing cells, but by the enzyme remaining from the previous transfer (at time 0). This leads me to wonder: are there other possible chemical species in this transfer volume that might impact the dynamics? For example, is it possible that the drug concentration in each transfer volume makes a nontrivial contribution to the overall drug concentration? If so, might the experiments with higher migration rates (i.e. more transfers or larger transfer volume) have systematically different drug concentrations than those with lower migration rates? Such an effect could be particularly strong if, for example, the drugs exhibit strong synergy or strong antagonism, so that even small changes in the concentration of one drug could dramatically alter the cellular response. I use drug concentration as an example, but the same applies to any environmental variable (e.g. pH or nutrient level) that might be present at sufficiently high concentration to impact growth and vary systematically for different migration rates. In short, what I'm getting at is this: how confident are you that the dominant effect of changing migration rates is truly due to the migration of *cells* to different habitats (rather than the tag-along "migration" of other molecules, etc., in the environment)?

Response 3: We believe that it is unlikely for chemical species (other than beta-lactamase) in the transfer volume to have a significant impact on the dynamics. First, we use a fairly large dilution factor of 100, which dilutes chemical species carried over from the previous day substantially. Moreover, we expect the antibiotics to be mostly deactivated over the course of 24 hours in our experiments, so the concentrations carried over are negligible. Further, since our *E. coli* strains are nearly identical genetically, their impact on environmental factors such as pH or nutrient concentration is similar. Hence, we expect the dynamics as a function of migration rate to be relatively insensitive to these parameters. Also, we note that the migration rate does not alter the effective dilution rate, since the same total volume of culture is transferred each day into fresh media. Finally, our model accurately captures the experimentally observed sequence of dynamical outcomes in spite of ignoring most of the aforementioned factors. In summary, we believe that the dominant effect of changing migration rates is due to the migration of cells, rather than the tag-along migration of molecules.

Comment 4: Many antibiotics are known to exhibit a strong inoculum effect (IE), where the initial density of cells modulates the inhibitory effects of the drug. You implicitly account for a cross-species IE in the model by modeling the impact of enzyme degradation. So, for example, the effect of Amp on the Chl-R cells will obviously depend on the density of Amp-R cells (or more directly, beta-lactamase enzyme) in the culture. In addition, a monoculture of Amp-R cells would be expected to exhibit a strong IE to ampicillin, and your model would capture that. But it's also possible that the Chl-R cells exhibit an IE when exposed to ampicillin, even when cultured alone, so that Chl-R growth in ampicillin depends on the initial density. The same goes for the Amp-R cells exposed to Chl. Because the densities of the subpopulations can change dramatically in this experiment, is it possible that the IE is playing a significant role here? To my knowledge, drugs similar to ampicillin are fairly often associated with an IE; sometimes this is due to production of beta lactamases—and again, you've covered that case in your model—but it can also be due to other mechanisms and/or result in increased drug efficacy at higher densities (see, for example www.ncbi.nlm.nih.gov/pmc/articles/PMC180193/ or <http://journals.plos.org/ploscompbiol/article?id=10.1371/journal.pcbi.1005098>). These other mechanisms could potentially impart a strong ampicillin IE (or “reverse IE”) to the Chl-R cells that wouldn't show up in your model. On the other hand, recent work in *E. coli* showed no IE with Chl (www.ncbi.nlm.nih.gov/pmc/articles/PMC3472685/), so it may not be so important for that drug.

Response 4: As indicated by the referee, enzymatic degradation of antibiotics is a known mechanism that leads to the inoculum effect (which we have studied in Artemova *et al.*, *MSB*, 2015). We have not observed a strong inoculum effect in ampicillin for strains that are sensitive to ampicillin. We therefore believe that the dominant effect driving our dynamics is enzymatic degradation.

Changes to the manuscript: We have now included the references cited by the reviewers (Ref. 42-44) in the revised manuscript. We have also added the following sentence (Line # 184-187) to the main text.

“While our mathematical model explicitly considers the inoculum effect due to enzymatic degradation of antibiotics, it does not include other potential sources of an inoculum effect (42-44), as they were not necessary to recapitulate the dynamics observed in experiments.”

Very minor points:

Comment 5: I would suggest that the authors (slightly) extend their discussion about the potential application of these results to the spread of antibiotic resistance, where cooperation between resistant and sensitive cells is increasingly recognized as an important contributor to population dynamics. In addition to cooperation, the current work highlights the importance of spatial structure and population heterogeneity for understanding how resistance spreads in complex communities. While these topics have been the focus of a number of recent papers, they are sometimes underappreciated in the study of resistance. I know resistance is not the focus of the current work, but I think it would nevertheless interest readers who study resistance.

Response 5: Following the reviewer's suggestion, we have expanded our discussion to include the potential application of our results to the spread of antibiotic resistance.

Changes to the manuscript: We have added new references (Ref. 53-55) as well as the following sentences (Line # 349-355) in the section 'Discussion':

“Collectively, our findings highlight the importance of spatial structure and its impact on bacterial population dynamics in the context of the evolution of antibiotic resistance (53, 54). It is evident that cross-protection can render a population resistant to multiple drugs without any strain evolving resistance to more than one drug. Furthermore, similar to how drug tolerance can promote the evolution of resistance (55), it is possible that cross-protection, particularly in combination with spatial structure and migration, could provide time for strains to evolve multidrug resistance *de novo*.”

Comment 6: The authors use both “recolonization” and “re-colonization”, depending on location.

Response 6: We have now rectified this inconsistency in the revised manuscript.

Changes to the manuscript: We have used “recolonization” uniformly throughout the manuscript.

Reviewer #2 (Remarks to the Author):

Comment 1: In general, the match between the theoretical model and the experimental observations is stunning, and the authors do a very good job describing their model and experiment. This work adds an interesting new facet to the potential for intermediate migration rates to enhance persistence, not by the common route involving recolonization but by an alteration of dynamics that enhances the longevity of the population.

Response 1: We thank the reviewer for their positive assessment and for appreciating the significance of one of our primary results.

Comment 2: Much of the work that this paper builds upon focuses on theoretical models or experiments where population dynamics are not subject to serial dilution effects. This is obviously a cornerstone of the approach needed to undertake the experimental work and I don't think that this takes away at all from the strength of this paper. I do wonder however, how much the findings rely on the fact that the system undergoes a substantial reset with each dilution. Of course, the period three cycles likely require the serial dilutions to be present, but I wonder how sensitive the transition to the period 4 cycles is on the dilution factor. Playing with this parameter in the model may also help to demonstrate the mechanism giving rise to the period 4 cycles.

Response 2: As anticipated by the reviewer, serial dilutions are indeed necessary to observe period-3 cycles (Yurtsev, Conwill, Gore, *PNAS* 2016). The following bifurcation diagram shows the population dynamics in a single well for varying growth-dilution schemes where the cycle time and dilution factor varied such that there is a constant death rate due to dilution (i.e. $d'=100^{(\Delta T/24)}$).

The antibiotic concentrations in the fresh media are $10 \mu\text{g/ml}$ ampicillin and $8 \mu\text{g/ml}$ chloramphenicol (the benign condition in this manuscript). The simulations ran for 200 growth-dilution cycles, and this figure shows the AmpR/ChIR ratio for the last 50 cycles. For shorter dilution cycles and smaller dilution factors, the two mutualists reach a stable ratio; as the environment becomes more periodic (longer dilution cycles and larger dilution factors), we see period doubling and eventually period-3 oscillations (starting at ~ 21.5 hr growth-dilution cycles).

Given that the growth-dilution scheme strongly influences the population dynamics in a single population, it is not surprising that the growth-dilution scheme strongly influences the dynamics in coupled populations as well. Below are bifurcation diagrams showing the population dynamics in two populations connected via migration, with a similar format to Figure 4C in the manuscript but with different growth-dilution schemes: the first panel shows a growth cycle of 16.5 hours (where there are period-2 oscillations in a single well); the second panel shows a growth cycle of 20.0 hours (where there is not a limit cycle oscillation in a single well); and the third panel shows a growth cycle of 22.5 hours (where there are period-3 limit cycles in a single well, similar to the 24 hour case). Simulations included 500 growth-dilution cycles at each migration rate starting from multiple initial AmpR/ChIR ratios, and the AmpR/ChIR ratio is shown for the last 50 cycles. These simulations further emphasize the rich population dynamics that can emerge as a result of migration, even in a relatively simple system.

Comment 3: Along this line of reasoning, I think the authors could improve on their description of the mechanistic basis for the change in dynamical behavior. The switch from a period 3 cycle to a period 4 cycle due to a change in migration rate is very novel and interesting and something that other models based on coupled oscillators have not shown. It is in fact hard to understand how increasing migration could effectively slow cycles down; passive migration has strongest effect in patches with low density and helps populations to increase quickly from the zenith of their cycle. For predator-prey oscillations this is often referred to as the ‘slow’ part of the cycle. My suspicion here is that somehow migration is acting in some alternative manner, to slow down the fast part of the cycle.

Response 3: In our system, the period-4 cycles emerge when the migration rate is high enough to delay the period-3 cycle, but not high enough to synchronize the two population patches. Specifically, the AmpR/ChIR ratio decreases twice and then increases once in the period-3 cycle. When the AmpR/ChIR ratio is at its intermediate point, migration from a neighboring well (that is out of phase) adds sufficient AmpR cells to dampen the decrease in the AmpR/ChIR ratio over the course of the day. Particularly because the AmpR/ChIR ratio changes over multiple orders of magnitude throughout the cycle, even a small migration rate can have a significant impact on the state of a population patch.

The following figure shows simulations of the cell densities and antibiotic concentrations over single growth cycles for the period-3 oscillations and for the period-4 oscillations (AmpR and Amp in red; ChIR and ChI in blue). The horizontal axis through the center of the figure represents AmpR/ChIR ratios: the three purple dots represent the AmpR/ChIR ratio for the period-3 limit cycle ($m = 0$; no migration) and the four green dots represent the AmpR/ChIR ratio for a period-4 limit ($m = 0.08$; intermediate migration rate). In both cases, the AmpR/ChIR ratio decreases (leftward) until it hops back upward (to the right). While the within-day dynamics in the first and last cycles are similar for both the period-3 and period-4 oscillations, the within-day dynamics for cycle 2 are in some sense repeated the next cycle in the case of the period-4 oscillation, suggesting that migration does indeed “delay” the second cycle by an extra day.

While the above description explains the mechanism behind why migration can cause the oscillation period to become longer, or “slow down”, it does not address whether or not this behavior is broadly relevant. In order to demonstrate that this behavior is not unique to our mathematical model, we turn to a much simpler model employing a coupled map.

Since daily dilutions impose discreteness on the population dynamics of our strains, we can compare the dynamics of our system with those observed in coupled nonlinear maps. A variety of coupled nonlinear maps relevant to ecology, such as the logistic and Ricker maps are known to undergo an inverse period doubling cascade at intermediate migration rates, which can lead to period-4 dynamics over an appreciable range of migration rates (Y. Ben Zion *et al.*, G. Yaari and N. M. Shnerb, *PLoS Comput Biol* 6(1): e1000643, 2010). As an illustrative example, we have considered a system of two coupled modified Ricker maps. In the absence of migration, the dynamics of each map are of the following form

$$x_{n+1} = x_n^3 \exp(-px_n)$$

The uncoupled maps exhibit period-3 oscillations for certain values of the parameter p , which correspond to a periodic window within a chaotic regime. Below, we have plotted two representative time series for this model demonstrating a change from period-3 dynamics at low migration rates (left panel, $m = 0.001$) to period-4 dynamics at intermediate migration rates (right panel, $m = 0.105$).

This transition is the result of an inverse period doubling cascade that leads to period-4 dynamics over a modest range of migration rates. In our mechanistic model as well, we observe inverse period doubling within the regime $0.05 < m < 0.06$. We therefore believe that the change from period-3 to period-4 dynamics in our system is the result of an inverse period doubling cascade, similar to what has been observed in a variety of coupled non-linear maps.

Changes to the manuscript: We have now added the following sentences to the main text (Line # 230-235) to explain the transition from period-3 to period-4 oscillations:

“The transition from period-3 to period-4 dynamics occurs via an inverse period doubling cascade (Fig. 4C, $0.05 \leq m \leq 0.06$). Such inverse period doubling cascades are commonly observed in discrete two-patch population dynamics models such as coupled logistic maps or Ricker maps (29). Although our model is based on ordinary differential equations, the daily dilution scheme imposes discreteness on the dynamics, and the observed bifurcation structure is analogous to that observed in coupled maps.”

Comment 4: From figure 3 panels c,d,e (or figure 4c) it really isn't clear why there is a difference in persistence times among the three parameter regimes due to the fact that the ratio of cell types is shown rather than absolute densities. I think showing the dynamics of the absolute densities would also be helpful so that differences in the dynamics (e.g. the minimum densities) would be obvious.

Response 4: Fig. 3C-E as well as Fig. 4C show data for the benign condition in which all populations survive over the entire duration of the experiments as well as simulations, and hence, there is no difference in persistence times among the three parameter regimes. In addition, for these benign conditions the bacterial cultures reach a similar density (within 30%) at the end of each day, meaning that there is no significant difference between fractions and absolute densities.

However, in the case of the harsh environment the population densities do exhibit significant fluctuations, and many populations eventually go extinct. Absolute population size data for the two strains in a harsh environment is available in Figs. S4 and S10.

Comment 5: Throughout the paper the authors use the term unsynchronized. I find this term to be imprecise. My preference is to use the terms synchronous to represent positively correlated patch dynamics, asynchronous to represent patches with no correlation, and antisynchronous for negatively correlated patches. This language has been adopted by a number of papers (although some authors argue that antisynchrony is just a special case of synchrony).

Response 5: Following the reviewer's recommendation, we have now used the terms asynchronous, synchronous, and antisynchronous to refer to the various types of dynamics observed in our experiments and simulations.

Changes to the manuscript: We have now replaced the term "unsynchronized" by "asynchronous", "in-phase synchronized" by "synchronous", and "out-of-phase synchronized" by "antisynchronous" throughout the manuscript.

Comment 6: It isn't clear what role the lag in growth has in the model, nor how the authors chose the lag value. Is this set by independent measurements, or was the parameter fit to best be able to describe the dynamics? Nowhere in the paper was the value of the lag presented.

Response 6: In the original manuscript, we presented the value of the lag time in Table S1 of the Supplementary Information. The value of the lag time (1 hour) and indeed, most other parameters, are identical to those used in our previous work (E. A. Yurtsev, A. Conwill, and J. Gore, *PNAS*, 113, 6236-6241, 2016). We have now clarified this point in the caption of Table S1. The value of the lag time chosen in the model is close to the value measured in experiments using similar strains (Yurtsev, Chao, *et al.*, *MSB*, 2013).

Changes to the manuscript: We have now explained our choice of parameter values in the caption of Table S1.

Comment 7: In addition, it isn't clear why the dynamics of A1 would depend only on the initial state of N1 rather than dynamically on the instantaneous value. Given that the model can only be analyzed numerically, this seems a rather odd simplifying assumption with no real benefit.

Response 7: In our previous work (Yurtsev, Chao, *et al.*, *MSB*, 2013), we measured significant beta-lactamase activity in the media transferred from the previous day, and in (Yurtsev, Conwill and Gore, *PNAS*, 2016) we observed that, while using the instantaneous density of ampicillin resistant cells, $N1(t)$ or the initial density, $N1(t=0)$ produced oscillatory dynamics, using $N1(t=0)$ produced robust period-3 oscillations similar to those observed in experiments. This observation suggested that the robustness of period-3 oscillations arises due to beta-lactamases carried over from the previous dilution cycle. We therefore retained this feature of the model for the present work. We have now mentioned this point in the main text.

Changes to the manuscript: We have added the following sentence (Line # 178-182) to the main text explaining our reason for using $N1(t=0)$ instead of $N1(t)$ in the model:

“These features of the model are supported by our previous findings that there is significant beta-lactamase activity in the media transferred from the previous day (41) and that the oscillatory population dynamics in our model of the mutualism most closely reflect those observed in experiments when we use the initial AmpR cell density to determine the ampicillin deactivation rate.”

Comment 8: In the low migration regime, is the antisynchronous state attractive, i.e. does the same dynamic setup regardless of the initial conditions or is it sensitive to initial conditions?

Response 8: In the low migration rate regime, dynamics are largely asynchronous. While populations starting with a very low initial cell density or extreme ratios of AmpR to ChIR cells may become extinct, most initial conditions lead to stable period-3 oscillations in the low migration rate regime. However, the phase of oscillation depends on initial conditions. In the intermediate migration regime, period-4 oscillations can coexist with period 3 oscillations in that different initial conditions could give rise to either type of oscillation. This can be seen in the bifurcation diagram in Figure 4C.

Comment 9: In the intermediate parameter range $0.02 < m < 0.06$ it is most likely that the dynamic is a form of quasiperiodic chaos since there is no obvious period doubling cascade taking place. This is typically generated by an interaction of oscillators with an irrational ratio of periods but I have to admit that I’m puzzled as to exactly how this type of dynamic sets up in this instance. It would be interesting to speculate on why the complex dynamics emerge here.

Response 9: In the benign environment (Fig. 4C), the chaotic dynamics are associated with an inverse period doubling cascade. Most of the irregular dynamics observed in the range $0.02 < m < 0.05$ are long

chaotic transients. In the figure above, we have expanded a small region of the bifurcation diagram shown in Fig. 4C to highlight the inverse period doubling cascade observed over $0.05 < m < 0.06$. For this bifurcation diagram, we simulated trajectories over 10,000 daily dilutions instead of the 1000 dilutions used for Fig. 4C, in order to obtain a more accurate picture of the asymptotic dynamics.

For the harsh environment (Fig. S5), it is quite possible that the chaotic dynamics emerge via the quasiperiodic route. The quasiperiodic route to chaos has been observed in a system of two logistic maps

with linear coupling (A. Lloyd, *J. Theo. Biol.*, 1995). In this system quasiperiodic dynamics are generated through a series of Hopf bifurcations as the coupling parameter is varied. The incommensurate frequencies necessary for quasiperiodicity are generated via these successive Hopf bifurcations. As the parameter is varied further, the system exhibits a number of complex dynamical phenomena including doubling of invariant tori, frequency locking, and chaos. While our system differs from the model studied by Lloyd, there are striking similarities between the attractors observed in our study and Lloyd's. These attractors can be visualized using Poincare recurrence plots for the log ratio of AmpR to ChIR cells, i.e. $x = \log(\text{AmpR}/\text{ChIR})$ (see figure below for representative attractors).

We note that some of the complex attractors (e.g. $m = 0.054$) we observe are not associated with positive Lyapunov exponents (Fig. S6 in the revised manuscript), which may indicate the presence of quasiperiodicity. Further, the complex limit cycles that we observe for certain migration rates (period-14 cycle at $m = 0.025$ in the figure above) may be associated with frequency locking, similar to that observed by Lloyd. While a detailed characterization of the routes to chaos is outside the scope of this study, we reiterate that the discreteness imposed by daily dilutions likely makes the dynamics of our system analogous to those observed in coupled maps. We have now mentioned these possibilities in the main text.

Changes to the manuscript: We have now included new references related to the observation of the quasiperiodic route to chaos in population dynamics models (Ref. 48-50) in the revised manuscript. We have also added the following sentences (Line # 316-322) to the main text:

“Since we do not observe period doubling, it is possible that the quasiperiodic route to chaos underlies the chaotic dynamics observed here. The quasiperiodic route has been observed in a system of two linearly coupled logistic maps (48) as well as other population dynamics models (49, 50). While a detailed analysis of the routes to chaos is outside the scope of the present work, we reiterate that the discreteness imposed by daily dilutions may render the dynamics of our system analogous to those observed in coupled nonlinear maps.”

Reviewers' Comments:

Reviewer #1:

Remarks to the Author:

The authors have satisfactorily addressed all of my comments. In my view, the additional analyses have improved an already strong manuscript. The results are compelling and novel, and this work will surely interest a broad range of readers, from ecologists to physicists.

Reviewer #2:

None